# Statistical Considerations for the Design and Analysis of Pragmatic Trials in Aging Research

**DOI:** 10.3390/geriatrics9030075

**Published:** 2024-06-04

**Authors:** Ashuin Kammar-García, Liliana Aline Fernández-Urrutia, Jorge Alberto Guevara-Díaz, Javier Mancilla-Galindo

**Affiliations:** 1Dirección de Investigación, Instituto Nacional de Geriatría, Mexico City 10200, Mexico; 2Lown Scholars in Cardiovascular Health Program, Departments of Global Health and Population and Epidemiology, Harvard TH Chan School of Public Health, Harvard University, Boston, MA 02115, USA; 3St. Luke School of Medicine, Alliant International University, Mexico City 11000, Mexico; fdzuuliliana@gmail.com; 4Faculty of Medicine, Autonomous University of Sinaloa, Culiacan 80019, Mexico; jorgeguevara.fm@uas.edu.mx; 5Institute for Risk Assessment Sciences, Utrecht University, 3584 Utrecht, The Netherlands; j.mancillagalindo@uu.nl

**Keywords:** pragmatic clinical trials, aged, statistical analysis, geroscience, sample size, secondary data analysis

## Abstract

Pragmatic trials aim to assess intervention efficacy in usual patient care settings, contrasting with explanatory trials conducted under controlled conditions. In aging research, pragmatic trials are important designs for obtaining real-world evidence in elderly populations, which are often underrepresented in trials. In this review, we discuss statistical considerations from a frequentist approach for the design and analysis of pragmatic trials. When choosing the dependent variable, it is essential to use an outcome that is highly relevant to usual medical care while also providing sufficient statistical power. Besides traditionally used binary outcomes, ordinal outcomes can provide pragmatic answers with gains in statistical power. Cluster randomization requires careful consideration of sample size calculation and analysis methods, especially regarding missing data and outcome variables. Mixed effects models and generalized estimating equations (GEEs) are recommended for analysis to account for center effects, with tools available for sample size estimation. Multi-arm studies pose challenges in sample size calculation, requiring adjustment for design effects and consideration of multiple comparison correction methods. Secondary analyses are common but require caution due to the risk of reduced statistical power and false-discovery rates. Safety data collection methods should balance pragmatism and data quality. Overall, understanding statistical considerations is crucial for designing rigorous pragmatic trials that evaluate interventions in elderly populations under real-world conditions. In conclusion, this review focuses on various statistical topics of interest to those designing a pragmatic clinical trial, with consideration of aspects of relevance in the aging research field.

## 1. Introduction

Pragmatic randomized controlled trials differ from explanatory randomized controlled trials in that the objective of pragmatic trials is to evaluate efficacy, typically in the context of usual patient care. In contrast, explanatory trials seek to assess the efficacy of an intervention, often under controlled conditions [1]. Despite observational studies being commonly used to approximate the effectiveness of an intervention, pragmatic trials are better at reliably answering questions of effectiveness since they can minimize confounding through randomization [2].

Although the distinction between explanatory and pragmatic trials could suggest a dichotomy exists between these types of trials, in practice, clinical trials can incorporate both explanatory and pragmatic elements. Therefore, the Pragmatic Explanatory Continuum Indicator Summary tool, second version (PRECIS-2 tool) [3] aids the evaluation and design of elements in the pragmatic–explanatory continuum of trials. In Figure 1, we provide an example of two different hypothetical trials in aging research with varying degrees of pragmatism, with an explanation of the design choices and PRECIS-2 scores provided in the Appendix A.

Pragmatic randomized controlled trials are increasingly being used in the aging research field due to the need of obtaining high-quality real-world evidence for interventions for the elderly, who tend to have low representation in trials [4]. Additionally, geriatric interventions are often complex in nature, which is the reason why pragmatic trials are useful designs for evaluating interventions [5]. Furthermore, pragmatic trials allow investigations in the context of regular clinical practice, with the advantages of being more accessible, being less resource-intensive, and placing minimal additional burden on participants [6]. 

Despite the multiple advantages of pragmatic trials for obtaining evidence for complex interventions in the elderly, there are several choices in the design of pragmatic trials that have important implications for the ability to obtain high-quality evidence, while minimizing costs. Guidance on such design choices is provided by the GetRealtrial tool [7,8]. Despite the existence of such tools, guidance and explanations on the rationale for the design and analysis of pragmatic trials from a biostatistician’s perspective remain scarce. Therefore, we sought to review the statistical considerations for the design and analysis of pragmatic trials, including available resources for the sample size calculation and analysis of pragmatic trials. In this review, we only cover statistical considerations from a frequentist approach and do not touch on Bayesian analyses, which have also been applied for the design and analysis of pragmatic trials. 

## 2. Study Unit and Randomization 

Although the individuals are the ultimate unit of interest in both explanatory and pragmatic trials, clusters are commonly used in pragmatic trials as the unit of randomization. A cluster refers to any level of grouping of individuals (i.e., patients who receive care from one single practitioner, a clinic or hospital, a jurisdiction, etc.). Cluster randomized trials (CRTs) allow for the estimation of the broad population effects of an intervention [9]. Randomization by clusters is also attractive in the context of pragmatic trials since they allow circumvention of logistical challenges of interventions delivered to a very large number of patients, among other reasons. 

A parallel cluster study in which different groups of individuals are assigned to receive an intervention or the comparator (i.e., placebo) without random assignment would be considered a quasi-experimental study and would have many sources of latent potential confounding. Fortunately, different randomization strategies [10] have been envisioned and successfully applied: Parallel randomized clusters: The clusters are randomized at the beginning of the study and remain unchanged until the end of the study;Parallel randomized clusters with a baseline period: In this design, observations are made for a period of time before randomization;Stepped wedge cluster randomized studies: In this design, all groups go from control to intervention at different times, and observations are obtained before and after the switches.

Hemming et al. [10] exemplify these designs with different studies carried out in diverse populations.

From a causal research perspective, the reason randomization is important in a conventional individual-level randomized study is that it allows for comparability of the prognosis of participants allocated to treatment groups [11]. Confounders are said to be randomly distributed between groups. Thus, the group to which participants are assigned serves as an instrumental variable [12] that can be used to approximate the effect of an intervention (assuming compliance with it). This is the principle of intention-to-treat [13] and the reason why the method of analysis should be Randomization-Based Inference [14], meaning that the principle of intention-to-treat should be followed. In this approach, subjects are evaluated considering the original group to which they were randomly assigned, and data elimination due to lack of information, treatment changes, use of other medications, or lack of adherence should be strongly avoided [3,15].

In pragmatic clinical trials, cluster randomization can often be recommended over individual randomization. Therefore, the number of clusters or the number of subjects per cluster should be determined a priori [16]. It is common to assume an equal number of subjects in each cluster (cluster size), leading to statistical analysis using hypothesis testing to compare means or proportions, depending on the type of dependent variable chosen. However, when cluster sizes are not equitable, the use of mixed effects models (also called random effects models) or generalized estimating equations (GEEs) is suggested [17].

The evaluation of missing data should be conducted to detect the presence of non-random patterns of unavailable data. Using imputation techniques may or may not be warranted, but it is imperative to assess if missing data exhibit a specific pattern, as non-random patterns could bias the results, leading to potentially incorrect interpretations [18].

## 3. The Dependent Variable

The type of dependent variable used in pragmatic clinical trials will guide statistical treatment, i.e., whether the variable is a continuous quantitative outcome or a dichotomous or ordinal categorical outcome. The choice of outcome variable should be made with caution because a variable that requires strict follow-up or subsequent clinic or hospital visits could interfere with “usual care” if the visit frequency differs from routine clinical care [15]. The chosen outcome variable should align with the pragmatic concept, reflecting usual clinical practice. Therefore, a continuous outcome (reduction in HbA1c, decrease in serum lipids, or fewer hospitalization days) can be commonly used to evaluate intervention effectiveness, as can a dichotomous outcome (achieving < 7 units of HbA1c, having an LDL < 150 mg/dL, or recovery from illness) [15].

It must be ensured that the choice of outcome represents the objective for which the usual treatment is utilized, but it should also be considered that a dichotomous variable may not fully represent the nature of the phenomenon that is intended to be studied or modeled in statistical terms. This is because a dichotomous variable leaves aside other potentially clinically relevant information to classify a single health status into two possible outcomes. A common outcome in many clinical studies is mortality, but dichotomizing a patient between “healthy” and “non-surviving” can hide relevant information for clinicians [19]. Using ordinal outcomes reduces such “hiding” of information and provides a clearer picture of the natural history of the disease (i.e., 1 = Healthy, 2 = Sick, 3 = Severely Ill, and 4 = Death). Ordinal outcomes have statistical properties more similar to quantitative variables, thus providing greater statistical power to detect clinically relevant differences [20] in situations where it may be difficult to observe due to the context (real-world settings) or population (older persons). An example of the use of ordinal outcomes in aging research is the trial by Spertus et al. [21], where the effect of two interventions on the health status of participants was investigated by using an ordinal scale outcome.

The most common study designs in pragmatic clinical trials or cluster trials are parallel designs. In these designs, the use of independent statistical tests (two-sample *t*-test, ANOVA, or χ^2^ test) is standard practice, whereas in crossover studies or designs where matching between clusters or individuals has been used, paired analyses (paired *t*-test, Friedman test, or McNemar test) have been employed [17]. In situations in which an ordinal variable is intended to be used as the outcome, ordinal logistic regression is a suitable method of analysis to estimate if an intervention performs better than the comparator for all outcomes (ordinal categories) at the same time [19]. Such “global” estimation of the effectiveness of an intervention may have more pragmatic relevance than binary outcomes for some studies. Thus, it is our opinion that future pragmatic trials in aging research could consider exploiting the properties of ordinal outcomes to make more pragmatic assessments of the global effectiveness of interventions over multiple health states, rather than focusing on single binary outcomes one at a time. However, we must emphasize that the choice of outcome is an individual decision that should be made for the purposes of individual trials, which is the reason why no single type of outcome should be taken as the best or become standard.

Since maintaining homogeneous cluster sizes is not always feasible, even in explanatory cluster-crossover trials [22], the use of mixed models and GEEs is highly recommended. However, their use less widespread than expected, leading to heterogeneity in the types of statistical analyses [23]. Mixed effects models and GEEs are longitudinal data analyses that allow estimation of the effect of an intervention on the outcome. Still, they differ in how they generate an estimation of the effect.

Mixed models allow for modeling the effects of fixed factors, which assume a constant effect, and random factors, which presuppose variability among subjects. They can be used when estimating the effect of an intervention considering the heterogeneity among the clusters to which subjects belong, and this heterogeneity can be modeled through a probability distribution. Estimates generated in mixed models are termed conditional estimates as the models provide a conditional estimate of the outcome given by the covariates or random effects [24,25].

## 4. Types of Statistical Models for Pragmatic Designs 

GEEs allow estimation of the average effect of a predictor variable across the entire study population; hence, they are termed population average models or marginal models. Such estimation of the effect is averaged across all clusters, making them suitable for estimating the effect of a predictor variable when the impacts of random factors are not of interest to the researcher. Therefore, GEEs do not require data distribution assumptions but require larger sample sizes for precise estimations [25,26].

The use of mixed models is more widespread for CRTs due to their ability to model different random effects. Table 1 presents the main mixed models, considering the intercept or slope as a random factor in the model. In this table, we provide a brief description of their use in CRTs, as well as the statistical model and code for use with the R statistical software. The dependent variable here is considered dichotomous, following previous recommendations regarding the use of dichotomous variables in pragmatic studies [27].

It is important to consider that the type of dependent variable (whether quantitative, dichotomous, or ordinal) can be modeled using linear mixed models (LMMs) or generalized linear mixed models (GLMMs). GLMMs, which depend on the distribution of the dependent variable, are modeled with different link functions (binomial, logit, Poisson, log–log, etc.). Regardless of the data modeling approach, it is crucial to verify the statistical assumptions of the models and compare between models using information criteria (AIC and BIC) when constructing models incorporating various variables [24]. One virtue of ordinal dependent variables is that even under violations of the proportional odds assumptions, a model assuming proportional odds may still be adequate provided it is the most parsimonious model compared to a proportional odds model or multinomial model (i.e., *impactPO* function in the *rms* R (≥4.1.0) package listed in Table 2). 

The way variability is modeled in an experiment can take various forms. In this review, we provide methodological guidance for the statistical design of a pragmatic clinical trial. Therefore, we suggest readers explore forums and delve into more profound literature concerning the application of such models across different software platforms. Additionally, it is vital to understand the requirements for data capture in data matrices, which differ from the conventional data matrix format where each row represents a different subject. Li F et al. [14] provide a comprehensive compilation of packages for developing such models in software like R, Stata, and SAS. In Table 2, we mention a small selection of R packages for the analysis of pragmatic trials. A full list of R packages for designing, monitoring, and analyzing randomized controlled trials can be consulted in the CRAN Task View for clinical trials [28].

## 5. Sample Size Estimation

The sample size calculation for pragmatic studies will depend on the chosen study design, specifically whether randomization of interventions is performed at the individual or cluster level. Sample size calculations have been described in multiple publications, and online calculators are available to estimate the required number of subjects based on whether the dependent variable is continuous or dichotomous [29]. However, in the case of cluster-based studies, an adjustment must be made for a correction factor known as the “variance inflation ratio” or “design effect”. This factor represents the multiplier by which the calculated sample size for individual randomization should be multiplied [22]. This design effect is calculated as follows: (1)$$D=1+\left(m-1\right)\rho ,$$
where m is the number of subjects per cluster, and ρ is the intracluster correlation coefficient $(s_{c}^{2}/s_{c}^{2}+s_{w}^{2}$), defined as the ratio of the variance of means between clusters ($s_{c}^{2}$) to the sum of the variance of subjects within the same cluster ($s_{w}^{2}$) and between clusters [30]. The calculation of the design effect in designs comparing two means uses ρ, whereas in designs comparing two proportions, the calculation of the cluster concordance index (κ) is employed [31]. It is important to mention that the calculation of the design effect assumes a homogeneous distribution of the number of subjects per cluster. Therefore, one may consider adjusting the sample size calculation assuming unequal cluster sizes through the calculation of the “coefficient of variation of cluster size” (cv), which can be executed by various methods explained in depth by Eldridge SM et al. [22].

As mentioned in the Dependent Variable section, equitable cluster sizes are not always estimated in pragmatic studies, and there is a desire to control for the effect of variation within clusters and between subjects. Hence, mixed models or GEEs are used, although these models serve to describe an effect size based on a different coefficient B, which is different from the classic effect sizes with mean or proportion differences, which are typically employed in classic sample size calculations. These more complex statistical models can be used even if the sample size calculation was based on a difference in statistics; however, it is preferable to calculate the sample size based on the estimation of a conditional (LMM-GLMM) or marginal (GEE) model. Therefore, the design effect should also be specific to these models. Li F et al. [14] compile various packages for sample size calculation in specific situations for different software (R, SAS, and STATA) for various types of CRTs. Likewise, Hemming K et al. [32] developed an online app for calculating sample sizes and statistical power for various CRT designs. In Table 2, we provide useful links to online calculators for sample size estimation for multiple scenarios.

## 6. Multi-Arm Study Sample Size

Throughout this paper, we have emphasized that pragmatic trials aim to test interventions in real-world situations. In some cases, there may be more than one standard of care, multiple promising new treatments, or various ways to implement an intervention. This is where multi-arm studies become relevant. The most common form of analysis for multi-arm studies involves comparing means between three or more groups using a general linear model (ANOVA family). For such comparisons, sample size calculation is performed considering an expected effect size (η^2^ or Cohen’s f) in the ANOVA model, statistical power (1−β), the alpha error probability (confidence level), and the number of groups to be included in the study [32]. However, this methodology estimates the sample size considering only the null hypothesis of the test (no group mean difference), so it does not consider multiple group comparisons (post hoc tests), which ultimately results in the lower statistical power of this calculation, leading to a higher risk of type 2 error. Adjusting the alpha error using the Holm–Bonferroni method (α/number of pairwise comparisons) can help provide a better estimate of the sample size. As previously mentioned, it is more common for the outcome used in usual care to be dichotomous rather than quantitative; therefore, comparing three or more proportions can be a pragmatic outcome. In this scenario, sample size calculation can be performed using formulas for comparing two proportions and adjusting the alpha error using the Holm–Bonferroni method. Grayling MJ et al. [33] developed a sample size calculator for multi-arm clinical trials for various types of variables and sequences and employed different multiple comparison correction methods. Additionally, it is important to note that adjustment for the design effect should also be considered if the sample size will be estimated for a CRT, once the multi-arm sample size is calculated.

The so-called “adjustment for losses” used in sample size calculations must be justified to avoid unnecessarily exposing more subjects to risk since, in a pragmatic nature, participants should be included in the analysis regardless of their follow-up losses or incomplete data. Finally, since prior data on effect sizes and, in the case of CRT, intracluster correlation are required to perform any sample size calculation, it is highly recommended that researchers report these statistics obtained in their study samples to assist future researchers in scaffolding their sample size calculations. Otherwise, they may be forced to conduct pilot studies, which could entail additional effort and expenses regarding the pragmatic clinical trial being conducted.

## 7. Secondary or Ancillary Analyses

The population included in any pragmatic study can be very heterogeneous, which can lead to the desire to compare the results of the primary outcome among subgroups of the sample to identify strata for which the intervention may be more effective [34]. This practice is common in clinical trials where secondary variables are sought to be evaluated beyond the original protocol due to possible hypotheses obtained during the main study or an attempt to assess effects among participant subgroups [35]. The issue with secondary or subgroup analyses is that they generally have fewer observations and hence less statistical power, increasing the risk of not detecting differences (type 2 error) or detecting them only by chance (type 1 error) [36].

Sample size calculation allows us to identify the minimum number of subjects needed to achieve sufficient statistical power to detect a difference between study groups on the primary outcome. Therefore, the statistical inferences we make in a study regarding secondary outcomes may be biased if a sample size was not calculated a priori for such a comparison [15]. It is under this premise that secondary analyses of clinical trials should be approached with caution, and the possibility of committing type 1 and 2 errors should be considered when a proper sample size calculation was not performed or when subgroup comparisons are overused [34]. If secondary analyses of a pragmatic clinical trial are to be conducted, efforts should prioritize the evaluation of outcomes relevant to clinical practice, while the use of surrogate markers is discouraged [15].

While the primary focus of a pragmatic clinical trial is always on evaluating the effectiveness of an intervention in a real-world setting, it is important to note that information on the safety of interventions is also collected [37]. Greater care must be taken regarding the method of safety data collection, as an excessive burden on healthcare providers can compromise the pragmatism of the study. It is suggested to use a combined strategy of data collection present in clinical records and case-form reports for serious adverse events [37]. In geriatrics, there is often a scarcity of studies dedicated to assessing the safety of medications in older adults, highlighting the imperative need for acquiring real-world evidence [38].

## 8. Methodological Challenges and Limitations

Although this review is mainly dedicated to providing guidance on the statistical considerations for conducting clinical trials, it is essential to mention that an adequate methodological design is crucial to bringing pragmatic clinical trials to an adequate conclusion. Thus, mention of the main limitations and methodological challenges of pragmatic clinical trials could help the reader to adequately specify the search for answers to their pragmatic questions regarding elderly populations. In general, pragmatic clinical trials have a series of challenges when trying to generate real-world evidence. The choice of the usual standard of care as well as the sites where the study will be conducted [39] are amongst these limitations. In aging research, nursing homes should be considered as possible research sites, and not only clinics or hospitals, since it has been seen that the application of interventions with older people when they are implemented in nursing homes has different results [40], and the reasons for this may be diverse both in the sites and in the populations served therein [41]. Therefore, the selection criteria for the participants are a challenge. Inclusion criteria must allow all those who may be candidates for usual care to be eligible, thereby generating an essential source of heterogeneity that can make the internal validity of the study lower. It has been suggested to include a large number of subjects if the intention is to carry out subgroup analysis of some characteristic of interest to researchers [34]. Furthermore, the obtention (or lack thereof) of informed consent is also a challenge to consider in pragmatic clinical trials, since the traditional way of requesting informed consent may not be viable in such studies. Kalkman et al. [42] discuss the existing alternatives for the obtention or waiving of informed consent in pragmatic studies. Finally, the collection and management of data is a challenge not only because of the amount of information to be collected, but also because of the excess workload that it can generate for the clinicians participating in the study. Meinecke AK et al. [18] review the alternatives that can avoid this extra burden on collaborators of pragmatic studies.

## 9. Conclusions

In this review, we have covered relevant aspects for the design and statistical analysis of pragmatic randomized controlled trials from a frequentist approach. The methodological design, the distribution of the dependent variable, the consideration of ordinal outcomes, the correct calculation of sample size, and the choice of the number of secondary analyses to be carried out are important statistical considerations that, alongside other choices in the design of pragmatic trials, are of utmost importance for the validity of the estimation of the effectiveness of interventions delivered in real-world conditions in elderly populations. This review should be considered as a compilation of relevant information and recommendations that can serve as a guide for those interested in the design and statistical analysis of pragmatic clinical studies in the aging research field.

## Figures and Tables

**Figure 1 geriatrics-09-00075-f001:**
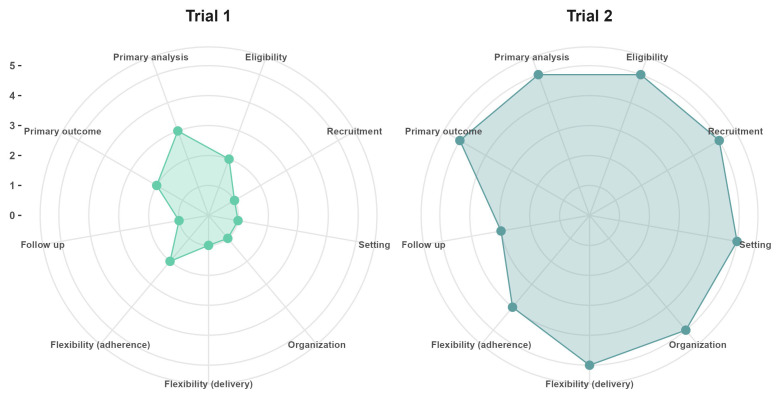
PRECIS-2 scores of two hypothetical pragmatic trials in aging research. Trial 1 (mean PRECIS-2 score = 1.6) refers to the situation in which a primary healthcare practitioner who is also a researcher at an academic research center wants to assess if a new drug is safe and capable of preventing secondary cardiovascular events in older adults after recovering from acute myocardial infarction, whereas Trial 2 (mean PRECIS-2 = 4.7) was designed after stakeholders commissioned a study to evaluate if implementing a new drug in all primary healthcare clinics of their jurisdiction will prevent secondary cardiovascular events under real-world conditions. Trial 1 had more explanatory components, whereas Trial 2 followed more pragmatic choices in the design.

**Table 1 geriatrics-09-00075-t001:** Types of mixed models.

Type of Model	Use in CRTs	Statistical Model	Basic R Code (*lme4* package)
Model with random intercept and fixed slope	It is useful for modeling a heterogeneous initial effect (intercept) among clusters or subjects, but with a homogeneous effect of the independent variable. It serves when assuming that members of a cluster have different initial values in the dependent variable.	$y_{\mathrm{ij}}=\beta_{0}+\beta_{1}x_{\mathrm{ij}}+b_{0j}+e_{\mathrm{ij}}$ where $y_{ij}$ is the dependent variable for subject i and group j; $\beta_{0}$ is the fixed intercept; $\beta_{1}$ is the fixed coefficient of the variable x; $b_{0j}$ is the random intercept effect for group j; and $e_{ij}$ is the error.	Model1 <- glmer (y ~ x + (1|cluster), family = binomial, data = data) (1|cluster): Indicates the random slope for each observation of the variable x and the random intercept for each cluster or subject.
Model with fixed intercept and random slope	It is useful for modeling that the effect of a dependent variable will be heterogeneous among the clusters or subjects, but that all subjects or clusters have similar values at the beginning of the study.	$y_{\mathrm{ij}}=\beta_{0}+\beta_{1}x_{\mathrm{ij}}+b_{1j}x_{\mathrm{ij}}+e_{\mathrm{ij}}$ where $y_{\mathrm{ij}}$ is the dependent variable for subject i and group j; $\beta_{0}$ is the fixed intercept; $\beta_{1}$ is the fixed coefficient of the variable x; $b_{1j}$ is the random slope effect for group j; and $e_{\mathrm{ij}}$ is the error.	Model2 <- glmer (y ~ x + (x|1), family = binomial, data = data) (x|1): Indicates the random slope for each observation of the variable x.
Model with random intercept and random slope	This model, known as a random effects model, is used to model the initial differences in the values of the dependent variable among clusters or subjects as well as the heterogeneous effect of the independent variable among clusters or subjects.	$y_{\mathrm{ij}}=\beta_{0}+\beta_{1}x_{\mathrm{ij}}+b_{0j}+b_{1j}x_{\mathrm{ij}}+e_{\mathrm{ij}}$ where $y_{\mathrm{ij}}$ is the dependent variable for subject i and group j; $\beta_{0}$ is the fixed intercept; $\beta_{1}$ is the fixed coefficient of the variable x; $b_{0j}$ is the random intercept effect for group j; $b_{1j}$ is the random slope effect for group j; and $e_{\mathrm{ij}}$ is the error.	Model3 <- glmer (y ~ x + (x|cluster), family = binomial, data = data) (x|cluster): Indicates the random slope for each observation of the variable x and the random intercept for each cluster or subject.

**Table 2 geriatrics-09-00075-t002:** Links to online resources for statistical analysis and sample size calculation of pragmatic trials.

Purpose	Link to the Resource
**R Packages for Statistical Analysis**	
*table1*: baseline characteristics	https://CRAN.R-project.org/package=table1 (access date: 21 May 2024)
*lme4:* linear mixed effects models	https://CRAN.R-project.org/package=lme4 (access date: 21 May 2024)
*nlme*: non-linear mixed effects models	https://CRAN.R-project.org/package=nlme (access date: 21 May 2024)
*rms*: regression modeling strategies	https://CRAN.R-project.org/package=rms (access date: 21 May 2024)
*mice*: imputation of missing data	https://CRAN.R-project.org/package=mice (access date: 21 May 2024)
*geeCRT*: bias-corrected generalized estimating equations for cluster randomized trials	https://CRAN.R-project.org/package=geeCRT (access date: 21 May 2024)
**Sample Size and Power Calculation**	
Cluster clinical trials	https://douyang.shinyapps.io/swcrtcalculator/ (access date: 21 May 2024)
Multi-arm trials	https://mjgrayling.shinyapps.io/multi-arm/ (access date: 21 May 2024)
Non-inferiority studies with binary outcomes	https://search.r-project.org/CRAN/refmans/dani/html/sample.size.NI.html (access date: 21 May 2024)
Non-inferiority studies with continuous outcomes	https://search.r-project.org/CRAN/refmans/epiR/html/epi.ssninfc.html (access date: 21 May 2024)
Studies with ordinal outcomes	https://search.r-project.org/CRAN/refmans/Hmisc/html/popower.html (access date: 21 May 2024)

## Data Availability

The original data and code presented in the study are openly available in GitHub at https://github.com/javimangal/PRECIS-2-example (access date: 20 May 2024).

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
