# Peer review of "Statistical Considerations for the Design and Analysis of Pragmatic Trials in Aging Research"

_geriatrics, 2024, doi:10.3390/geriatrics9030075_

Round 1

Reviewer 1 Report

Comments and Suggestions for Authors

This is an interesting report about the importance of statistical approaches of the pragmatic trials in aging. I enjoyed reading it. As a whole, the manuscript is well-written, clear, and concise.

I recommend to add additional information which are beneficial to the paper to strengthen it.

1-    Pragmatic trials have been criticized for having the several problems. I suggest to add a paragraph at the end of your manuscript about method limitations. You can refer to many articles (such as Series: Pragmatic trials and real-world evidence. Clinical journal of epidemiology (2017)).

2-    Can you list R packages which can be used to analyze the data?

Minor issues:

1-    There are some punctuation errors:

-         In the tittle: add a space between the words “trial” and “in” ; In lines:  57 and 59

2-    Please provide the full name of the PRECIS tool, then you can use the abbreviation 

Author Response

This is an interesting report about the importance of statistical approaches of the pragmatic trials in aging. I enjoyed reading it. As a whole, the manuscript is well-written, clear, and concise.

R=Thank you very much for your comment, and for the time invested in carrying out your review. In the current version, we attach an extra paragraph on the use of ordinal outcomes and the benefits that this brings in clinical studies on aging. I hope you like the review version where they were carried out. Likewise, the corrections suggested by both reviewers.

I recommend to add additional information which are beneficial to the paper to strengthen it.

1-    Pragmatic trials have been criticized for having the several problems. I suggest to add a paragraph at the end of your manuscript about method limitations. You can refer to many articles (such as Series: Pragmatic trials and real-world evidence. Clinical journal of epidemiology (2017)).

R=Thank you very much for your suggestion. We are attaching a new section to the end of the document: “8. Methodological challenges and limitations”, where we describe the main methodological challenges and limitations of pragmatic studies. We see this suggestion as very useful for giving a broader overview to readers, thus helping to better prevent problems arising from these studies, which helps to provide better answers to pragmatic questions.

2-    Can you list R packages which can be used to analyze the data?

R=Thank you for your suggestion. In the document attached to Table 2, we place various R packages that can be used for various statistical analysis purposes.

Minor issues:

1-    There are some punctuation errors:

-         In the tittle: add a space between the words “trial” and “in” ; In lines:  57 and 59

 R= Thank you very much for the observation. We modifed the text in the title and fixed the errors in lines 57 and 59.

2-    Please provide the full name of the PRECIS tool, then you can use the abbreviation

R= Thank you for the observation; we add the full name of the tool.

Reviewer 2 Report

Comments and Suggestions for Authors

This paper describes statistical considerations for the design and analysis of pragmatic trials (trials in which the objective is to evaluate efficacy, usually in the context of usual patient care in aging research). Overall, the guidelines build well on extant statistical methodology and are well chosen. Here are some items to attend to in a revision.

First, the text of the paper needs a careful edit of the grammar. Here are a couple of examples:

Line 63: "... for the obtention of evidence..."; "obtention" must be misspelled as it is not a commonly used word; given the intent of what you are trying to convey, "for obtaining evidence" would be better phrasing.

The acronym CRT appears to be first used in line 152 and in the heading of a column of Table 1. Like other acronyms used in the text, it should be placed in parentheses after the full English words that it references.

Second, it a few places in the text, it will make your paper more readily accessible to readers who are interested in pragmatic trials if you would elaborate some referents that are mentioned with references to other publications. An example is "parallel randomized clusters, parallel randomized clusters with a baseline period, and stepped wedge cluster randomized studies". The inclusion of brief verbal definitions of these study designs would be appropriate.

Comments on the Quality of English Language

As indicated in my comments to the authors, the text of the paper needs a careful edit of the grammar. 

Author Response

This paper describes statistical considerations for the design and analysis of pragmatic trials (trials in which the objective is to evaluate efficacy, usually in the context of usual patient care in aging research). Overall, the guidelines build well on extant statistical methodology and are well chosen. Here are some items to attend to in a revision.

R= Thank you very much for your comment and for the time you invested in carrying out your review. In the current version of the review we added a few lines with comments from reviewer 1 and other statistical considerations such as the use of ordinal outcomes.

First, the text of the paper needs a careful edit of the grammar. Here are a couple of examples:

Line 63: "... for the obtention of evidence..."; "obtention" must be misspelled as it is not a commonly used word; given the intent of what you are trying to convey, "for obtaining evidence" would be better phrasing.

R= Thank you for the correction. We made changes to the manuscript and asked a native English speaker to give an extensive review.

The acronym CRT appears to be first used in line 152 and in the heading of a column of Table 1. Like other acronyms used in the text, it should be placed in parentheses after the full English words that it references.

R=Thank you very much for the observation, we placed the acronym after the first mention with the complete words; this was placed on line 83 of revised version.

Second, it a few places in the text, it will make your paper more readily accessible to readers who are interested in pragmatic trials if you would elaborate some referents that are mentioned with references to other publications. An example is "parallel randomized clusters, parallel randomized clusters with a baseline period, and stepped wedge cluster randomized studies". The inclusion of brief verbal definitions of these study designs would be appropriate.

R= Thank you very much for your suggestion. We modified the wording in the indicated sentence, giving a brief description of the designs and a reference to a study where different examples can be seen. Likewise, in this version of the review, we place a paragraph explaining the statistical considerations of the use of ordinal outcomes, and at the end of the paragraph, we place an example of how these have been used in aging research.

Round 2

Reviewer 2 Report

Comments and Suggestions for Authors

The revisions to this paper have been responsive. The manuscript has been improved accordingly. Minor editing: It would be good to emphasize the contributions of the paper to the existing literature in the Abstract and Conclusions.

One word in the text to change: "aggrupation" in line 81; I did not find this word in the dictionary; change it to "grouping".

Comments on the Quality of English Language

The English grammar has been improved and now is acceptable.

Author Response

The revisions to this paper have been responsive. The manuscript has been improved accordingly. Minor editing: It would be good to emphasize the contributions of the paper to the existing literature in the Abstract and Conclusions.

R=Thank you very much for your review and your suggestion, we attach two sentences in the conclusion of the abstract and in the conclusion of the text, where we mention what our article contributes to the current scientific literature on aging research.

One word in the text to change: "aggrupation" in line 81; I didn't find this word in the dictionary; change it to "grouping".

R=Thank you very much for your correction, we modify the word as you suggest